



# Enhancing characterization of organic nitrogen components in aerosols and droplets using high-resolution aerosol mass spectrometry

Xinlei Ge[1,2], Yele Sun[1,3], Justin Trousdell[1], Mindong Chen[2], Qi Zhang[1]

[1] Department of Environmental Toxicology, University of California at Davis, One Shields Avenue, California 95616, United
States
[2] Jiangsu Key Laboratory of Atmospheric Environment Monitoring and Pollution Control (AEMPC), Collaborative Innovation
   Center of Atmospheric Environment and Equipment Technology (AEET), School of Environmental Science and Engineering,
   Nanjing University of Information Science & Technology, Nanjing 210044, China
[3] State Key Laboratory of Atmospheric Boundary Layer Physics and Atmospheric Chemistry, Institute of Atmospheric Physics,
Chinese Academy of Sciences, Beijing 100029, China

*Correspondence to*: Qi Zhang (dkwzhang@ucdavis.edu)

**Abstract.** This study aims to enhance the understanding and application of the Aerodyne high-resolution aerosol mass spectrometer (HR-AMS) for comprehensive characterization of organic nitrogen (ON) compounds in aerosol particles and atmospheric droplets. To achieve this, we analyzed seventy-five N-containing organic compounds, representing a diverse range
of ambient ON types, including amines, amides, amino acids, N-heterocycles, protein, and humic acids. Our results show that ON compounds can produce significant levels of $NH_x^+$ and $NO_x^+$ ion fragments, which are typically recognized as ions representative of inorganic nitrogen species. We also discovered the presence of $CH_2N^+$ at $m/z = 28.0187$, an ion fragment that is rarely quantified in ambient datasets due to substantial interference from air-related $N_2^+$. As a result, we determined that an updated calibration factor of 0.79 is necessary to accurately quantify ON content using aerosol mass spectrometry.

20       We also assessed the relative ionization efficiencies (RIEs) for different ON species and found that the average RIE of ON compounds ($1.52 \pm 0.58$) aligns with the commonly used default value of 1.40 for organic aerosol (OA). Moreover, through a careful examination of the HR-AMS mass spectral features of various ON types, we propose fingerprint ion series that can aid the ON speciation analysis. The presence of $C_nH_{2n+2}N^+$ ions is closely linked with amines, with $CH_4N^+$ indicating primary amines, $C_2H_6N^+$ suggesting secondary amines, and $C_3H_8N^+$ representing tertiary amines. $C_nH_{2n}NO^+$ ions (especially for n
values of 1-4) are very likely derived from amides. The co-existence of three ions, $C_2H_4NO_2^+$, $C_2H_3NO^+$, and $CH_4NO^+$, serves as an indicator for the presence of amino acids. Additionally, the presence of $C_xH_yN_2^+$ ions indicates the occurrence of 2N-heterocyclic compounds. Notably, an elevated abundance of $NH_4^+$ is a distinct signature for amines and amino acids, as inorganic ammonium salts produce only negligible amounts of $NH_4^+$ in HR-AMS.

29       Finally, we quantified the ON contents in submicron particles ($PM_1$) and fog waters in Fresno, California and $PM_1$ in New
York City (NYC). Our results revealed the substantial presence of amino compounds in both Fresno and NYC aerosols, whereas concurrently collected fog waters contained a broader range of ON species, including N-containing aromatic heterocycle (e.g., imidazoles) and amides. These findings highlight the significant potential of employing the widespread HR-



AMS measurements of ambient aerosols and droplets to enhance our understanding of the sources, transformation processes, and environmental impacts associated with ON compounds in the atmosphere.

**1. Introduction**

Nitrogen-containing organic species (ON) have been ubiquitously observed in atmospheric particles and aqueous phases (e.g., fog/cloud droplets, and rainwater). These compounds can constitute a significant portion (~ 30%) of the total airborne nitrogen (Cape et al., 2011) and contribute to up to 25% of the total nitrogen (N) deposition flux (Jickells et al., 2013; Kanakidou et al., 2016). However, our understanding of atmospheric ON is generally limited compared to inorganic N species such as ammonium, nitrate and nitrite. This disparity arises from the diverse nature of ON, which encompasses a wide range of compounds with varying carbon numbers, functional groups, and physicochemical properties.

ON compounds, originating from multiple emission sources and formed through diverse chemical processes, exhibit notable spatial and temporal variations in the atmosphere. The presence of airborne ON carries significant implications for regional air quality, earth's climate, terrestrial and aquatic ecosystem, and human health (Cape et al., 2011; Cornell et al., 1995; Liu et al., 2017). Among these compounds, aliphatic amines have been found to play a significant role in new particle formation and growth (Smith et al., 2010; Almeida et al., 2013; Yao et al., 2018), and contribute to secondary organic aerosol formation (Murphy et al., 2007; Yu et al., 2017; Song et al., 2017). Furthermore, ON species have been demonstrated to significantly influence the optical and hygroscopic properties of aerosols (Powelson et al., 2014; Lee et al., 2013; Rovelli et al., 2017) , thereby influencing cloud formation and atmospheric water cycles. These observations highlight the importance of studying ON compounds and their behaviors in the atmosphere.

Bulk ON, which represents the nitrogen content within organic compounds, is typically quantified by subtracting the inorganic nitrogen (IN) content from that of the total nitrogen (TN): ON = TN – IN. IN is the combined nitrogen amount in ammonium, nitrate and nitrite (Zhang and Anastasio, 2001).  Measurement of TN involves converting N-containing species into inorganic ions (e.g., $NO_3^-$, $NO_2^-$, and $NH_4^+$) or nitrogen gases (e.g., $NO_x$ and $NH_3$) using chemical, photochemical, or high-temperature oxidation approaches (Cornell et al., 2003; Zhang and Anastasio, 2003b; Jickells et al., 2013). However, quantifying ON using the differing methods can introduce significant uncertainties, due to factors such as incomplete transformation of ON compounds and aggregation of measurement errors (Cornell et al., 2003; Cape et al., 2011). The latter is especially important when ON is minor compared to IN in the sample. Moreover, ON measurements have traditionally focused on the water-soluble fraction (WSON). However there are observations that water-insoluble ON (WION) can constitute a large fraction of the total ON and, sometimes may be more important than WSON (Miyazaki et al., 2011; Russell et al., 2003). Furthermore, most studies investigating atmospheric ON have relied on samples collected over periods of hours to days. The limited time resolution of these measurements hampers the ability to capture the rapid evolution processes of ON species in the atmosphere.



The speciation of atmospheric ON has been analyzed using various techniques, including ion chromatography
(Vandenboer et al., 2011; Verriele et al., 2012; Parworth et al., 2017; Liu et al., 2021; Place et al., 2017), gas chromatography
coupled with different detectors (ÖZel et al., 2011; zel et al., 2009; Akyüz, 2007), infrared (IR) and nuclear magnetic resonance
(NMR) spectroscopy (Herckes et al., 2007), liquid chromatography with mass spectrometry (LC-MS) (Ruiz-Jimenez et al.,
2012; Samy and Hays, 2013; Samy et al., 2011; Ye et al., 2017), high-resolution electrospray ionization mass spectrometry
(ESI-MS) (Laskin et al., 2009; Altieri et al., 2012; Rincon et al., 2012; Shi et al., 2020), aerosol mass spectrometry (Junninen
et al., 2010; Huang et al., 2012; Setyan et al., 2014; Zhou et al., 2016; Ge et al., 2014; Xu et al., 2017; Huang et al., 2021; Yu
et al., 2019), chemical ionization mass spectrometry (Yao et al., 2018; Zheng et al., 2015; Yao et al., 2016; Yu and Lee, 2012),
and   nano-secondary ion mass spectrometry (Li et al., 2016). It is important to note that no single technique can
comprehensively capture the full spectrum of ON species. However, certain mass spectrometric techniques may provide
broader coverage (Cape et al., 2011). A variety of ON species have been detected in the atmosphere, including amines, amino
acids, urea, amides, nitriles, organic nitrates, nitro-compounds, and N-heterocyclic compounds, and more. However, when
considered individually, these identified ON classes often contribute only a small fraction to the overall ON pool. For example,
amino compounds, including amines, amino acids, peptides and proteins, were found to constitute less than 20% of the total
ON in the Central Valley of California – a region known for intense agricultural emissions of these compounds (Zhang and
Anastasio, 2001; Zhang et al., 2002). Moreover, in some instances, ON compounds were only qualitatively characterized
without quantitative information. These limitations restrict a comprehensive understanding of the chemistry of atmospheric
ON compounds and their environmental impacts.

The Aerodyne high-resolution time-of-flight aerosol mass spectrometer (HR-AMS) is a highly promising tool for
characterizing ON in atmospheric condensed phases. It utilizes thermal vaporization and electron impact (EI) ionization
techniques to quantify aerosol components and has been widely used for real-time measurements of non-refractory components
in submicron aerosols ($PM_1$) with fast time resolution (Zhang et al., 2020; Decarlo et al., 2006). An important feature of the
HR-AMS is its ability to differentiate fragment ions with the same integer mass-to-charge ratio (*m/z*) but slightly different
exact masses. This capability allows for the determination of the elemental compositions of the detected ions (Decarlo et al.,
2006) and the average atomic ratios, such as oxygen-to-carbon (O/C), hydrogen-to-carbon (H/C), and nitrogen-to-carbon
(N/C), for the bulk organic aerosol (OA) (Aiken et al., 2008; Ma et al., 2021). These ratios are particularly valuable for
estimating the total ON content present in the aerosol samples.

While the speciation analysis of ON using the HR-AMS mass spectra (MS) can be challenging due to the extensive
fragmentation of parent molecules (Drewnick, 2012), it is still possible to identify different types of ON by leveraging unique
spectral fingerprints, such as those observed for aliphatic amines (Murphy et al., 2007; Sun et al., 2011), N-heterocycles
(Hawkins et al., 2018; Kim et al., 2019), and organonitrates (Farmer et al., 2010). Furthermore, the HR-AMS offers a new
perspective for source identification of ON in ambient air. Specifically, the highly time-resolved HR-AMS data, typically
collected at intervals of 2-5 minutes during field studies, offer valuable insights into the co-variation of ON species with other
known components and permit the application of multivariate factor analysis techniques to differentiate components that



exhibit similar patterns of behavior. A recent study has successfully applied positive matrix factorization (PMF) on the N-containing ions in an HR-AMS field dataset, and determined ON factors that can facilitate the inference of specific sourced or

formation processes (Qi et al., 2022).

To achieve a comprehensive characterization of atmospheric ON using HR-AMS, it is necessary to optimize and validate the technique. While the interpretation of OA behaviors has often relied on the O/C and H/C ratios (Aiken et al., 2008; Canagaratna et al., 2015), the N/C ratio has received limited attention. To address this knowledge gap, our study conducted analyses on a large set of ON standards, encompassing diverse chemical types likely present in the atmosphere. The HR-AMS

spectra obtained were meticulously analyzed to characterize N-containing ion fragments and establish the relationship between the N/C ratios measured by HR-AMS and the nominal N/C values of the compounds. In addition, we carefully investigated mass spectral features of the ON standards and proposed speciation analysis protocols. Furthermore, we evaluated the effectiveness and applicability of the proposed method by examining three ambient HR-AMS datasets. Our results demonstrate that the optimized HR-AMS technique and the proposed analysis protocols carry great promise at enhancing our ability to

investigate the sources, transformation processes, and environmental impacts of atmospheric ON.

## 2. Experiments and data analysis

A comprehensive set of seventy-five pure ON standards were analyzed by the HR-AMS, and detailed information regarding these standards can be found in Table S1 of the Supplementary Material. These standards, which include 27 amines (No. 1-27), 6 amides (No. 28-33), 27 amino acids (No. 34-59 and 71), 4 nitro-compounds (No. 60-63), 7 N-containing

heterocycles (No. 64-70), 1 protein (No. 72), and 3 humic-like substances (No. 73-75), were mostly procured from Sigma-Aldrich with purities exceeding 98%. In addition, to probe the relative ionization efficiencies (RIEs) of different ON species, we analyzed 18 mixtures containing sulfate (in the form of ammonium sulfate) (Fisher Chemicals, > 99%) and individual ON standards in a 1:1 mass ratio (Table S2). All solutions were prepared using purified water (>18.2 MΩ·cm) obtained from a Milli-Q system (Millipore, USA).

The procedures for HR-AMS analysis of liquid samples were detailed in previous work (Sun et al., 2010; Ge et al., 2017; O'brien et al., 2019; Niedek et al., 2023). Briefly, the aqueous solutions were nebulized using high purity argon. A diffusion dryer was employed to dehumidify the aerosol before it was introduced into the HR-AMS. The HR-AMS was operated at a vaporizer temperature of ~ 600 °C and was switched between the highly sensitive V-mode and the high mass resolution W-mode (m/Δm ~5000). To account for any potential contamination or background signals, a blank measurement was conducted

by aerosolizing Milli-Q water between every two samples, following the same procedure.

The HR-AMS MS were analyzed using the standard AMS data analysis software, namely SQUIRREL v1.46/v1.51H and PIKA v1.06/v1.10H (http://cires.colorado.edu/jimenezgroup/ToFAMSResources/ToFSoftware/index.html) written in Igor Pro (Wavemetrics, Portland, USA). During data post-processing, measures were taken to eliminate interferences caused by air components on organic fragments (e.g., $N_2^+$ at $m/z$ 28, $O_2^+$ at $m/z$ 32). Given that the relative humidity values recorded during



the measurements remained consistently very low (<5%), the influence of particle-bound water was deemed negligible. Thus, the MS analyses were conducted using the directly measured signals of $O^+$, $OH^+$, $H_2O^+$ and $CO^+$.

     HR-AMS data acquired from the online measurements of $PM_1$ during July-August 2009 in NYC (Sun et al., 2011), and during January 2010 in Fresno, CA (Ge et al., 2012b; Ge et al., 2012a) were used in this study. Additionally, a set of 11 fog water samples collected during January 2010 in Fresno (Kim et al., 2019) were measured and analyzed following the ON

standard analysis procedure described above. The analytical technique developed in this work was applied on these datasets to assess its effectiveness and applicability.

### 3. Results and discussion

### 3.1. N/C calibration factor

     Accurate determination of elemental ratios using HR-AMS spectra requires calibrations due to the loss of neutral

fragments and the inability to detect very small $m/z$ ions such as $H^+$. The calibration factor for N/C reported in Aiken et al. (2008) was established based on 27 ON standards, mainly consisting of amino acids. The N/C ratios derived from HR-AMS spectra for these standards agree well with the nominal values (slope = 0.96; $r^2$ = 0.95) and the estimated uncertainty associated with this calibration was 22%. In this study, we re-evaluated the N/C calibration factor using a considerably larger dataset comprising 75 ON standards that represent a broad range of N-containing functional groups (see Table S1). The new calibration

plot (Figure 1a) shows a similar slope (0.99 ± 0.03, $r^2$ = 0.84), albeit with a higher uncertainty of 32% compared to the previous calibration. This increased uncertainty is partially attributed to the large positive biases observed for a few primary amines (further details provided in Section 3.4). In addition, the calibration factors for O/C (0.75) and H/C (0.93) (Figures 1c and 1d), determined based on the ON standards in this study, exhibit good agreements with those reported by Aiken et al. (2008) (0.75 for O/C and 0.91 for H/C).

However, it should be noted that the N/C calibration factors reported here (0.99) and in Aiken et al. (2008) (0.96) include contributions from $NH_x^+$ and $NO_x^+$ ions. Yet, these ions are typically assigned to inorganic ammonium and nitrate, respectively, during the analysis of ambient AMS data. Consequently, adjustments should be made to the N/C correction factors to ensure accurate estimation of ON content in ambient aerosols. Table 1 shows that $NH_x^+$ ions are consistently observed in the MS of various types of ON standards, contributing approximately 10% to the total N content on average. $NO_x^+$ ions are particularly

significant in nitro-compounds, contributing on average 16.2 (±8.3)% to the total N. A previous study conducted on organic nitrates also demonstrated the predominant occurrence of $NO_x^+$ ions (mostly $NO^+$ at $m/z$ 30) compared to the much less abundant $C_xH_yO_zN_p^+$ fragments observed in the HR-AMS mass spectra (Farmer et al., 2010). Based on the standards analyzed in this study, excluding the contributions from $NH_x^+$ and $NO_x^+$ would lead to an overall reduction of ~ 13% in the N/C calibration ratio. In addition, the $CH_2N^+$ ion ($m/z$ = 28.0187) is an important fragment in most ON compounds, particularly

amines. However, due to interference from the adjacent $N_2^+$ ion, it is necessary to exclude the $CH_2N^+$ ion in the analysis of ambient HR-AMS data. By excluding this ion, the N/C calibration ratio would further decrease by ~7%.



Considering the exclusion of $NH_x^+$, $NO_x^+$, and $CH_2N^+$ ions, we have derived a new N/C calibration factor of 0.79 (Figure 1b). We highly recommend using this value to determine the N/C ratio for ambient samples. However, it is important to note that even with this new ratio, there may still be underestimations of the ON content in cases where aerosols/droplets contain a significant amount of urea or organic nitrates. This is because a large fraction of the N mass (generally > 50%) is present in $NH_x^+$ for urea and in $NO_x^+$ for organic nitrates.

**3.2 Relative ionization efficiencies (RIEs) of different ON species**

The RIEs of ON species were determined by analyzing the HR-AMS spectra of 18 mixtures containing ON species and sulfate in 1:1 mass ratio. The measured ratios between the total organic signal and the sulfate signal varied from approximately 0.3 to 2, with an average of 1.33 (Figure S1). Assuming that the aerosol composition measured by HR-AMS corresponds to the solute composition in the original mixtures, these ratios represent the RIEs of different ON compounds relative to sulfate. In this study, since the RIE of sulfate relative to nitrate (based on mainly $NO^+$ and $NO_2^+$) was found to be 1.14, the average RIE of ON species relative to nitrate was determined to be $1.52 \pm 0.58$. It should be noted that the obtained result from the analysis of 18 mixtures aligns well with the average RIE of $1.6 \pm 0.5$ reported by Price et al. (2023), which was based on a study involving only 3 compounds (4-nitrocatechol, isosorbide mononitrate and triammonium citrate). This RIE value (1.52) is ~8.6% higher than the widely used default value of 1.4 for the quantification of total OA (Canagaratna et al., 2007). However, considering the inherent measurement uncertainties, such as variations in volatility leading to deviations in HR-AMS-determined particle compositions from the original mixture ratios, and the potential influences of impurities in the solutions on the measured mass of ON and thus the resulting RIE, we recommend maintaining the default RIE of 1.4 for the quantification of ON in ambient aerosols and droplets.

**3.3. Quantification of ON in ambient aerosols/droplets**

The mass concentration of ON can be determined based on HR-AMS measurements using the following equations:

$$OC = OM_{mass} / (OM/OC) \quad\quad (1)$$
$$ON = OC \times (N/C) \times (14/12) \quad\quad (2)$$

where OC represents the mass concentration of carbon present in OA, $OM_{mass}$ denotes the measured total OA mass. The OM/OC (organic mass to organic carbon ratio) and N/C (calculated using the new correction factor of 0.79) can be derived from HR-AMS elemental analysis.

For liquid samples, such as aqueous extracts of collected filter samples, fog, cloud, and rain water, the ON concentrations can be determined in ppm or $mg\cdot L^{-1}$ using the OC content measured by a total organic carbon (TOC) analyzer, typically representing the water-soluble OC (WSOC) in ppm or $mg\cdot L^{-1}$. The corresponding WSON concentration can be calculated as follows:

$$WSON = WSOC \times (N/C) \times (14/12) \quad\quad (3)$$

where the factor (14/12) is the ratio of the molar masses of nitrogen and carbon, respectively.



195       In the above calculations, accurate quantification of ON relies on the accuracy of N/C determination, which can be influenced by the need to estimate $H_xO^+$ (mainly $H_2O^+$, $OH^+$, $O^+$) and $CO^+$ signals for ambient datasets (Aiken et al., 2008). The $H_xO^+$ signals do not directly affect the N/C ratio, and hence have no impact on the WSON calculation in Eq. 3. However, the current estimation of $H_xO^+$ ($H_2O^+ = 0.225\ CO_2^+$, $OH^+ = 0.25\ H_2O^+$, $O^+ = 0.05\ H_2O^+$) for ambient OA (Aiken et al., 2008) can introduce uncertainties in the O/C and H/C ratios, thereby influencing the accuracy of the OM/OC ratio. If Eq. 1 is used to calculate the OC content, an overestimation of the OM/OC ratio would lead to a lower OC and an underestimation of ON

in Eq. 2. The estimation of $CO^+$ ($= CO_2^+$; (Aiken et al., 2008)) can also affect the elemental ratios and ON quantification, albeit to a lesser extent compared to the $H_xO^+$ ions. These uncertainties can be minimized, though not entirely eliminated, by conducting measurements on HEPA-filtered ambient air during field campaigns.

      Additionally, in the HR-AMS spectra of ambient samples, ON ions are typically observed at substantially lower intensities compared to adjacent ions, and are often located at the edges or between $C_xH_y^+$ and $C_xH_yO_z^+$ ions. For instance, $C_xH_yN_1^+$ ions are often found between $C_xH_{y-2}O_1^+$ and $C_{x+1}H_{y+2}^+$ ions, while $C_xH_yO_1N_1^+$ ions are located between $C_xH_{y-2}O_2^+$ and $C_{x+1}H_{y+2}O_1^+$

ions (refer to Figure S2 for two examples at *m/z* 42 and 44). Therefore, accurate determination of ON ions requires careful optimization of the peak shape and peak width parameters prior to performing peak fittings. This optimization helps reduce uncertainties associated with fitting of ON ions, particularly those with low signal intensities. Furthermore, caution should be given when fitting ON ions with high *m/z* values (e.g., >100 amu) since the mass resolution of HR-AMS may not be sufficient

to unambiguously distinguish ON ions from neighboring high *m/z* isobaric ions.

### 3.4. HR-AMS mass spectral features of ON compounds

      Detailed analyses of the HR-AMS spectra of all tested ON standards (Figure S3) reveal unique fragmentation patterns associated with specific functional groups. Aliphatic amines, characterized by the strong electron-donating ability of the nitrogen atom, predominantly undergo α-cleavage, resulting in the formation of ion series of $C_nH_{2n+2}N^+$, e.g., $CH_4N^+$ (*m/z*

30.0344), $C_2H_6N^+$ (*m/z* 44.0500), $C_3H_8N^+$ (*m/z* 58.0657), $C_4H_{10}N^+$ (*m/z* 72.0813), $C_5H_{12}N^+$ (*m/z* 86.0970), depending on the molecular structures (Mclafferty and Turecek, 1993). Notably, primary amino compounds with the $-CH_2-NH_2$ functional group, such as ethylamine, ethanolamine, and glycine (Figure 2a), exhibit the most abundant peak of $CH_4N^+$ (*m/z* 30.0344). In the case of butylamine, iso-butylamine and pentylamine, the $CH_4N^+$ ion can constitute >60% of the total signals in their respective MS, leading to significant positive biases in the N/C ratios determined by HR-AMS, as shown in Figures 1a and 1b.

Additionally, secondary elimination from initial α-cleavage may contribute significantly to the formation of $C_nH_{2n+2}N^+$ ions for secondary and tertiary amines (e.g., diethylamine and triethylamine; Figure 2a).

      α-cleavage can be an important fragmentation pathway for amides as well, causing the loss of alkyl groups and the formation of ion series of $C_nH_{2n}NO^+$, such as $CH_2NO^+$ (*m/z* 44.0136), $C_2H_4NO^+$ (*m/z* 58.0293), $C_3H_6NO^+$ (*m/z* 72.0449), and $C_4H_8NO^+$ (*m/z* 86.0606) (Mclafferty and Turecek, 1993). Particularly, the $CH_2NO^+$ ion (sometimes also $CHNO^+$) can be

generated from the amide group ($-CO-NH_2$), as observed for urea, pyrazinecarboxamide, and L-asparagine (Figure 2b).



Additional, cleavage of the C(O)-N bond can be significant for certain amides, resulting in the formation of $C_3H_3O^+$ from acetaminophen and $C_7H_5O^+$ from benzamide (Figure 2b).

Amino acids containing the functional group -CH(NH$_2$)-COOH commonly produce significant signals of $C_2H_4NO_2^+$ ($m/z$ 74.0242) resulting from α-cleavage. Subsequent losses of -OH and -CO from the $C_2H_4NO_2^+$ ion lead to the formation of

$C_2H_3NO^+$ ($m/z$ 57.0215) and $CH_4NO^+$ ($m/z$ 46.0293), respectively. This pattern is observed in amino acids such as serine, valine, cysteine, leucine, and tyrosine as depicted in Figure 2c.

2N-heterocyclic compounds show characteristic $C_xH_yN_2^+$ peaks in their spectra (Figure 2d). For example, imidazole and nitro-imidazole produce $C_xH_yN_2^+$ ions at $m/z$ 67.0296/68.0374 ($C_3H_3N_2^+$/$C_3H_4N_2^+$) while histine generates $C_4H_5N_2^+$/$C_4H_6N_2^+$ at $m/z$ 81.0452/82.0531. The imidazolyl compounds also produce ion series of $C_{3+n}H_{5+n}N_2^+$ (n>1). On the other hand,

compounds containing the pyrazinyl group show notable $C_{4+n}H_{4+n}N_2^+$ series, with $C_4H_4N_2^+$ at $m/z$ 80.0374 representing pyrazinecarboxamide.

Nitro-compounds containing the -NO$_2$ group (e.g., nitrophenol, 2-nitrobenzaldehyde and 2-nitrobenzyl alcohol) display much higher $NO^+$/$NO_2^+$ ratio (averaging 8.64 for the 4 compounds tested in this study). This ratio is similar to those observed for organic nitrates (i.e., RONO$_2$) (Farmer et al., 2010). In comparison, the $NO^+$/$NO_2^+$ ratio in pure ammonium nitrate is much

lower, which is 2.69 in this work, ~2.7 in Fry et al. (2009) and 2.4 in Bruns et al. (2010). In addition, although not measured in this study, nitriles have been shown to generate principal ion series of $C_nH_{2n-1}N^+$ and $C_nH_{2n-2}N^+$ (Mclafferty and Turecek, 1993).

In the HR-AMS spectra of ON compounds, the contributions of molecular ions are generally low, averaging around 3.3% of the total signals. The relationship between the compounds' molecular weights and the abundance of molecular ions is not

straightforward (Figure S4). However, for compounds with stable structures including benzene or heterocyclic rings, a high abundance of molecular ion peak is typically observed. For example, pyrazole and imidazole exhibit a $C_3H_4N_2^+$ peak at $m/z$ 68.0374, 1,3-phenylenediamine shows a $C_6H_8N_2^+$ peak at $m/z$ 108.0687, 4-aminophenol displays a $C_6H_7NO^+$ peak at $m/z$ 109.0528, and nicotinamide exhibits a $C_6H_6N_2O^+$ peak at $m/z$ 122.0480. It is worth noting that a few simple and low molecular weight aliphatic amines, such as methylamine, dimethylamine, and trimethylamine, and urea - a simple amide, show relatively

high contributions from molecular ions in their HR-AMS spectra as well.

### 3.5. Speciation analysis of ON in ambient aerosols and droplets

The analysis of ON compounds or classes in AMS spectra of ambient samples presents challenges due to their low concentrations, the complexities and uncertainties involved in assigning ion fragments to potential parent molecules. However, by examining the spectral features as discussed earlier, we can effectively address these challenges. To provide a

comprehensive overview, we have compiled the mass contributions of different N ion categories to the total ON mass for different types of ON compounds and summarized them in Table 1.

With the exception of amides, the majority of ON mass originates from $C_xH_yN_1^+$ ions, accounting for over 60% of the total mass. This is particularly true for amines and amino acids. Considering the atmospheric abundances of amines (Ge et al.,



2011b, a), significant presence of $C_xH_yN_1^+$ ions strongly suggest the presence of amines. Moreover, $C_nH_{2n+2}N^+$ ions may

indicate the existence of aliphatic amines. To evaluate this further, we calculated the fractional contributions of three specific

ions – $CH_4N^+$, $C_2H_6N^+$ and $C_3H_8N^+$ (left panel of Figure 3) – in the HR-AMS MS of different ON classes. It is evident that

these three ions are only abundant in amines rather than in other types of N-containing species. Furthermore, a close

examination of the distribution of these ions among different types of amines (right panel of Figure 3) reveals that $CH_4N^+$ is

mainly associated with primary amines (e.g., methylamine, ethylamine, butylamine, pentylamine and ethanolamine), $C_2H_6N^+$

is more likely linked with secondary amines (e.g., dimethylamine and diethylamine), and $C_3H_8N^+$ is mainly produced by

tertiary amines (e.g., trimethylamine and triethylamine).

       Another intriguing observation is that only amines and amino acids can generate significant $NH_4^+$ ($m/z$ 18.0344) signal

compared to other ON standards (Figure 4). Conversely, the HR-AMS spectra of pure ammonium nitrate, ammonium sulfate

and ammonium chloride (Figure S6) do not produce the $NH_4^+$ ion. This disparity is likely due to the cleavage of C-N bond,

accompanied by hydrogen migration to the $NH_x^+$ (x=0, 1, 2) fragments. Importantly, this finding strongly indicates that $NH_4^+$

can serve as an HR-AMS marker for the identification of amino compounds, including both amines and amino acids.

       In addition, the contribution of $C_xH_yN_1O_1^+$ ions to the ON mass is generally small (<6%), except for amides, where it can

reach ~26% (Table 1). This observation suggests that the abundance of $C_xH_yN_1O_1^+$ ions, particularly $CH_2NO^+$, can serve as an

indicator of the presence of amides. Furthermore, we propose that the co-occurrence of $C_2H_4NO_2^+$ ($m/z$ 74.0242), $C_2H_3NO^+$

($m/z$ 57.0215), and $CH_4NO^+$ ($m/z$ 46.0293) can act as tracer ions for amino acids. Furthermore, an enrichment of $C_xH_yN_2^+$ ions

suggests the presence of certain 2N-heterocyclic compounds, such as imidazoles and pyrazines (Kim et al., 2019; Hawkins et

al., 2018). Finally, a high $NO^+/NO_2^+$ ratio is likely a signature indicating the abundance of nitro-compounds/organic nitrates

(Farmer et al., 2010; Fry et al., 2009).

       Since AMS uses standard 70 ev electron ionization, the obtained spectra are expected to be comparable with the EI spectra

available in the NIST database (National Institute of Standards and Technology Standard Reference Database 1A, NIST 11,

Software version 2.0g). Figure S5 shows the correlation coefficients (Pearson's $r^2$) between the AMS-measured spectra

(summed in unit mass resolution) and the NIST spectra for the 75 pure ON standards. The level of spectral similarity varies

among different types of ON compounds. Higher correlations are observed for amines (average $r^2$=0.70) and amides (average

$r^2$=0.63), while amino acids exhibit low correlations (average $r^2$=0.23). The $r^2$ values for N-heterocycles and nitrocompounds

are 0.57 and 0.50, respectively. Although an increase in molecular weight generally tends to decrease the spectral similarities,

the influence of molecular weight is not particularly evident (Figure S5).

       A more detailed analysis reveals that the AMS spectra of aromatic compounds with ring structures exhibit close

agreements with the corresponding NIST spectra (e.g., 1,3-phenylenediamine, 4-aminophenol, and 2-picolinic acid in Figure

S7). Conversely, spectra of aliphatic molecules (long-chain or branched) show less similarity and contain more signals towards

low $m/z$ range compared to the NIST spectra (e.g., tri-n-amylamine and lysine in Figure S7). Additionally, AMS spectra of

oxygenated compounds tend to deviate more significantly from their NIST counterparts than other compound types. This

divergence can be attributed to additional thermal energy provided by the 600°C AMS oven, resulting in more extensive



fragmentation compared to the NIST spectra. Indeed, previous investigations by our group have demonstrated that employing a lower AMS vaporizer temperature ($T_{vap}$) (e.g., 250°C) can reduce fragmentation, enhance the signals of parent molecular
ions and/or fingerprint fragments, and overall increase the resemblance to the corresponding NIST spectra (Ge et al., 2014). Docherty et al. (2015) also showed that the OA MS changes substantially with variations in $T_{vap}$, and utilizing a lower $T_{vap}$ can improve the analysis of some reduced OA species (with more $C_xH_y^+$ ions). Therefore, during field deployments (or offline laboratory studies), it would be valuable to periodically acquire AMS spectra at lower $T_{vap}$ in addition to the standard 600°C spectra, as they may provide richer chemical information for compound speciation.

**3.6. Characterization of ON in ambient aerosols and fog waters**

Using the new N/C calibration factor (0.79), we determined the N/C ratios of NYC $PM_1$, Fresno $PM_1$ and fog waters and summarized them in Figure 5. On average, Fresno $PM_1$ exhibited a higher N/C ratio compared to NYC (0.019 vs. 0.015), although both ratios were significantly lower than that of fog organics (0.078). The mean ON mass concentrations were estimated to be 70 ng·m$^{-3}$, 120 ng·m$^{-3}$ and 2.2 mg·L$^{-1}$ for the three sample sets, respectively. The ON levels observed in NYC
and Fresno $PM_1$ are relatively low compared to those observed in a forest site in southern China (0.2~1.1 μg m$^{-3}$) (Yu et al., 2020) and in Hong Kong (0.24 ± 0.09 μg m$^{-3}$) (Li et al., 2022; Yu et al., 2021), but are similar to the ON level observed in a recent study conducted in summertime Nanjing, China (0.08~0.14 μg m$^{-3}$) (Xian et al., 2023).

The distribution of ON ions in NYC $PM_1$, Fresno $PM_1$ and fog water are further shown in Figure 6. Across all spectra, $C_xH_yN_1^+$ ions dominated, accounting for 84%, 84% and 66% of the total ON mass respectively, indicating that amines are a
major ON component in both cities. Notably, significant $C_nH_{2n+2}N^+$ ions are observed, suggesting the presence of aliphatic amines, which aligns with previous findings (Zhang and Anastasio, 2003a, 2001). The presence of $C_xH_yNO_1^+$ ions likely indicates the presence of amides.

Comparatively, the HR-AMS spectra of Fresno fog waters also exhibited prominent $C_xH_yN_2^+$ ions, likely originating from imidazoles formed through aqueous-phase reactions (De Haan et al., 2009a; De Haan et al., 2009b; Kim et al., 2019). In
addition, appreciable signals of $C_2H_4NO_2^+$ (*m/z* 74.0242), $C_2H_3NO^+$ (*m/z* 57.0215), and $CH_4NO^+$ (*m/z* 46.0293), indicative of amino acids, are observed in fog waters but are negligible in $PM_1$, consistent with the findings reported in Zhang et al. (2002), where the quantification of amino compounds were achieved through derivatization HPLC analysis. The enrichment of amino acids in fog waters compared to $PM_1$ can be attributed to a significant fraction of amino acids originating from proteinaceous matter mainly associated with coarse mode particles, such as soil dust and pollen. While these amino compounds may evade
effective detection by the AMS in real time, they can be scavenged by the fog droplets.

**4. Method limitations**

In the preceding sections, we have explored the successful optimization of the HR-AMS methodology for the quantification and chemical characterization of ON species in atmospheric aerosols and droplets. Nonetheless, it is important to acknowledge





certain limitations associated with this method. Firstly, it is important note that previous studies have shown bimodal size
distributions of ON in marine aerosols (Violaki and Mihalopoulos, 2010; Cornell et al., 2001; Miyazaki et al., 2010), with a
significant contribution from supermicron particles ($PM_{>1}$). For example, the average concentrations of WSON were found to
be $5.5 \pm 3.9$ and $11.6 \pm 14.0$ nmol m$^{-3}$ for coarse ($PM_{1.3-10}$) and fine ($PM_{1.3}$) particles, respectively, at a remote marine location
in the Eastern Mediterranean (Violaki and Mihalopoulos, 2010). However, the HR-AMS has limitations in measuring ON in
coarse particles in ambient air due to constraints in the transmission efficiency of the aerodynamic lens. Secondly, the AMS's
thermal vaporization at ~600°C is not capable of vaporizing refractory materials, potentially leading to underestimation of ON
levels by excluding refractory ON species. Note that the contribution of refractory ON to TN is still not well known. Thirdly,
the current method for analyzing liquid samples is applicable to only WSON. Future investigations may focus on using HR-
AMS to analyze the water-insoluble ON (WION), for example, by measuring samples extracted with organic solvents (Jiang
et al., 2022; Chen et al., 2017). Lastly, while specific ion fragments have been identified that are likely associated with different
ON functional groups, the extensive fragmentation caused by the 70ev electronic ionization in HR-AMS introduces inherent
uncertainties in such assignments. The incorporation of information from soft ionization mass spectrometry techniques (Wang
et al., 2019; Song et al., 2022; Mao et al., 2022) in combination with AMS could greatly enhance the speciation analyses of
ON species, reducing uncertainties and further improving our understanding of their composition.

## 5. Conclusions

340       This study focuses on the development of methodologies using HR-AMS for quantification and characterization of ON
species in ambient aerosols and aqueous droplets. Through extensive analysis of HR-AMS spectra from 75 ON standards, we
reach several important conclusions. Firstly, we show that ON compounds can generate significant $NH_x^+$ and $NO_x^+$ ion
fragments, which are commonly assigned to inorganic nitrogen species in ambient AMS analysis. Additionally, although the
presence of $CH_2N^+$ ($m/z = 28.0187$) is common in the AMS spectra of ON compounds, it is often overlooked due to interference
from air-related $N_2^+$ in ambient datasets. Secondly, the average RIE for ON was determined to be 1.52 ($\pm$ 0.58), which aligns
well with the default RIE value of 1.4 for total OA quantification. By considering these factors, we recommend a new N/C
calibration factor of 0.79 for ON quantification.

      Furthermore, we examined the high-resolution mass spectra (HRMS) of various ON compounds to identify chemical
fingerprint ions that can aid in the speciation of ON components in ambient HR-AMS datasets. Our findings reveal distinct
ion patterns associated with different ON types. Specifically, $C_nH_{2n+2}N^+$ ions serve as tracer ions for amines (particularly
$CH_4N^+$, $C_2H_6N^+$ and $C_3H_8N^+$ are markers of primary, secondary and tertiary amines, respectively). The presence of $C_nH_{2n}NO^+$
ions (especially for n=1-4), is a strong signal for the presence of amides. Tracer ions for amino acids include $C_2H_4NO_2^+$,
$C_2H_3NO^+$ and $CH_4NO^+$. The $C_xH_yN_2^+$ ion series are characteristic of 2N-heterocyclic compounds. It is worth noting that $NH_4^+$
ion can be produced from amines and amino acids, but not from inorganic nitrogen species. Additionally, we compared the
HR-AMS and NIST database spectra for the ON species. The level of similarity varied significantly among different ON type.



Pearson's correlation coefficients ($r^2$) for amines, amides, N-heterocycles, nitro compounds and amino acids were found to be 0.70, 0.63, 0.57, 0.50 and 0.23, respectively.

Lastly, we applied the methodologies described above to analyze three HR-AMS datasets, which included online measurements of $PM_1$ and offline analysis of fog waters. The fog waters in Fresno exhibited significantly higher N/C ratios (average = 0.078) compared to the $PM_1$ samples (0.019 in Fresno and 0.015 in NYC). Regarding ON constitution, both Fresno and NYC $PM_1$ contained significant amounts of amino compounds, whereas Fresno fog waters exhibited a broader range of ON species, including N-containing aromatic heterocycles (e.g., imidazoles) and amides.

Moving forward, our aim is to extend this approach to multiple HR-AMS datasets collected from diverse locations worldwide. By doing so, we anticipate gaining a more comprehensive understanding of the chemical characteristics, sources, and processes of atmospheric ON. This broader application will contribute to advancing our knowledge regarding airborne organic nitrogen compounds and their role in atmospheric chemistry on a global scale.

*Code and data availability.* All datasets, including HR-AMS mass spectra, are available upon request from Qi Zhang (dkwzhang@ucdavis.edu.cn) and Xinlei Ge (caxinra@163.com).

*Supplement.* The supplementary material related to this article is available online.

*Author contributions.* QZ conceived and designed the experiments, while XG and YS conducted the experiments. XG, YS, JT and QZ analyzed the HR-AMS spectra of the standards and the ambient data. XG and QZ wrote the paper with input from all authors.

*Competing interests.* The contact author has declared that none of the authors has any competing interests.

*Acknowledgements.* This work was funded by the U.S. Department of Energy Office of Science Atmospheric System Research Program (Grant #DE-SC0022140), the U.S. National Institute of Environmental Health Sciences Core Center (Grant P30 ES023513), and the Natural Science Foundation of China (91544220, 21976093), and the Jiangsu Natural Science Foundation (BK20150042).

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

# Tables

Table 1. Average mass contributions (%) from major nitrogen-containing ion categories to the total organic nitrogen (ON) mass of different types of standards.

| Categories | Amines | Amides | Amino Acids | Nitro-compounds | N-Heterocycles | Protein | Humic Acids |
|---|---|---|---|---|---|---|---|
| | $N=27$ | $N=6$ | $N=27$ | $N=4$ | $N=7$ | $N=1$ | $N=3$ |
| $C_xH_yN_1^+$ | **84.4±14.1** | 48.1±20.8 | **76.8±12.5** | 60.8±13.8 | 69.8±19.3 | 75.5 | **81.3±3.7** |
| $C_xH_yN_2^+$ | 2.9±7.7 | 11.0±11.5 | 1.9±5.3 | 3.7±5.5 | **11.0±15.6** | 3.7 | 1.1±0.2 |





| | | | | | | | |
|---|---|---|---|---|---|---|---|
| $C_xH_yO_1N_1^+$ | 2.2±5.6 | **25.6±11.9** | 4.2±4.0 | 5.1±1.9 | 5.6±7.5 | 5.9 | 16.0±3.2 |
| $C_xH_yO_2N_1^+$ | 0.26±0.97 | 0.64±1.44 | 3.0±4.4 | 1.5±0.9 | 2.8±3.5 | 0.3 | 1.2±0.6 |
| $NO_x^+$ | 0.20±0.45 | 0.90±1.03 | 0.97±1.58 | **16.2±8.3** | 1.3±2.2 | 1.1 | - |
| $NH_x^+$ | **9.8±9.7** | **12.6±18.5** | **12.9±8.7** | **10.4±7.8** | **9.5±15.4** | **13.3** | - |






# Figures

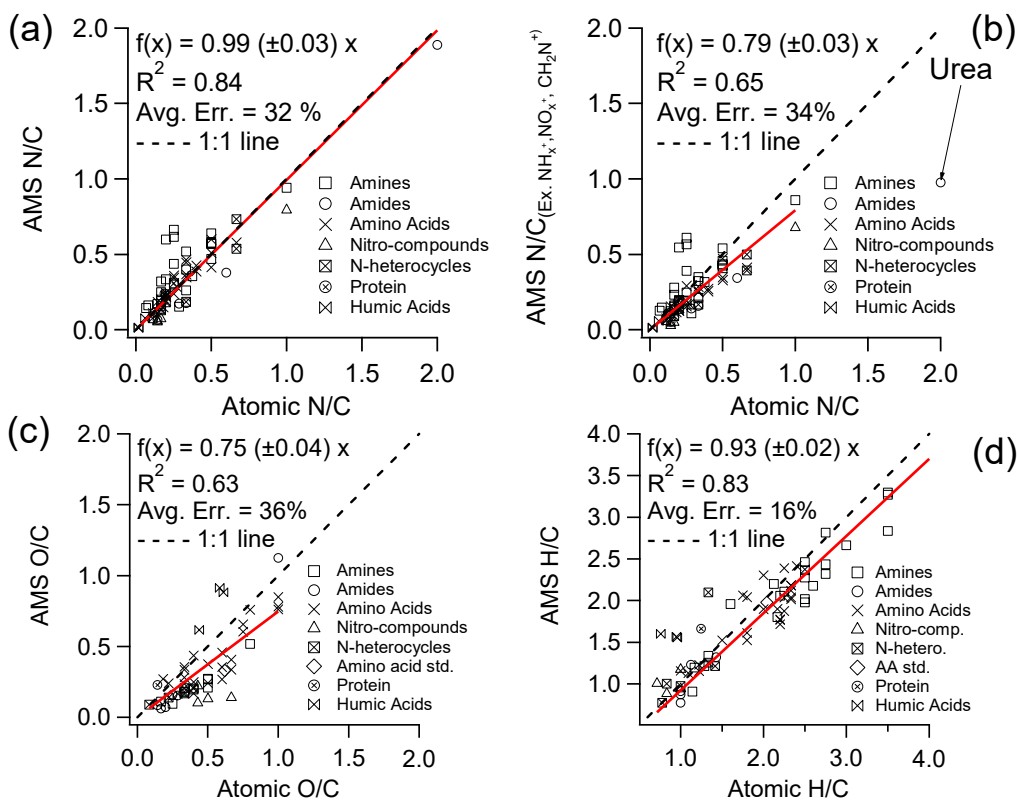

**Figure 1. Atomic N/C ratios from the elemental analysis of 75 ON standards versus the nominal values by (a) including and (b) excluding ion categories of $NH_x^+$, $NO_x^+$, and $CH_2N^+$, and the atomic ratios of (c) O/C and (d) H/C from the same set of ON standards versus nominal values (the linear regressions are forced to zero. Urea is not included in the linear regression in (b); Table S1 details the standards used here).**




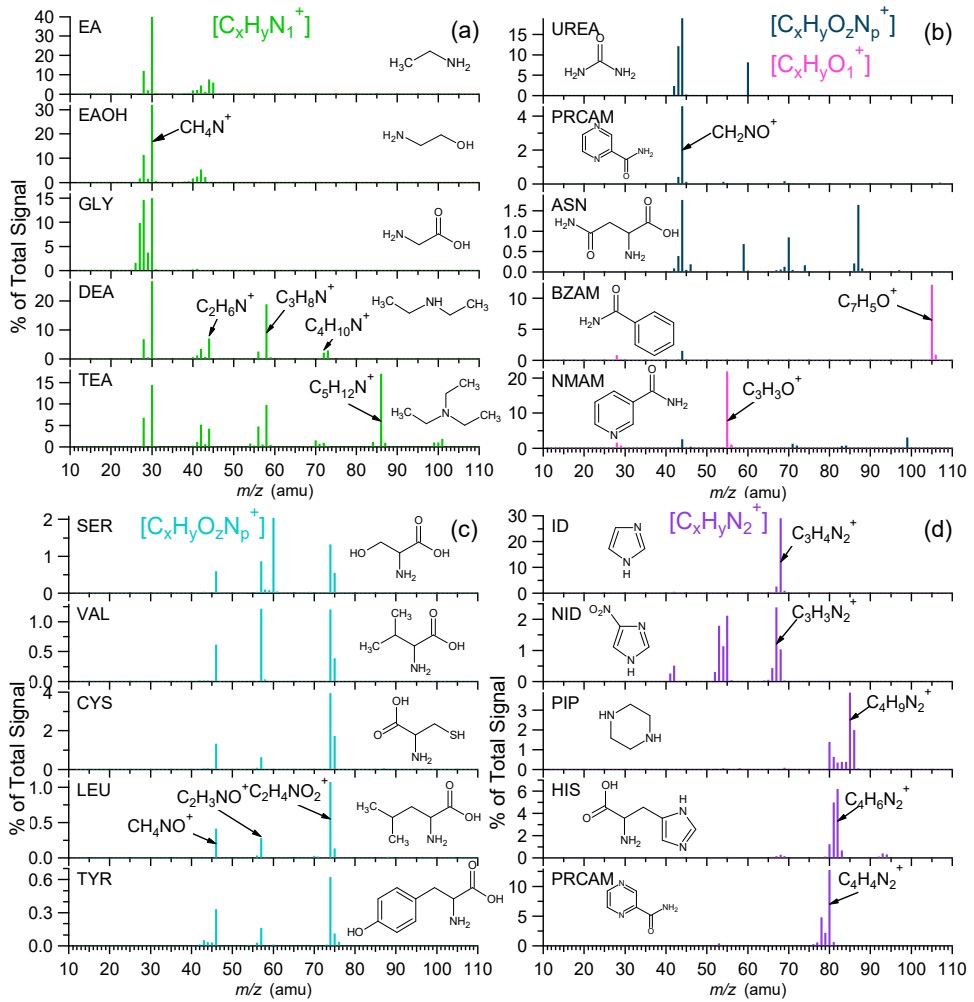

**Figure 2. Characteristic spectral peaks of selected ON standards with different functional groups of (a) amines (R-NH₂, R₂-NH, R₃-N), (b) amides (R-CO-NH₂), (c) amino acids (R-CH(NH₂)-COOH), and (d) 2N-heterocycles. EA: Ethylamine; EAOH: Ethanolamine; GLY: Glycine; DEA: Diethylamine; TEA: Triethylamine; UREA: urea; PRCAM: Pyrazinecarboxamide; ASN: Asparagine; BZAM: Benzamide; NMAM: N,N'-Methylenebisacrylamide; SER: Serine; VAL: Valnine; CYS: Cysteine; LEU: Leucine; TYR: Tyrosine; ID: Imidazole; NID: Nitro-Imidazole; PIP: Piperazine; HIS: Histidine.**



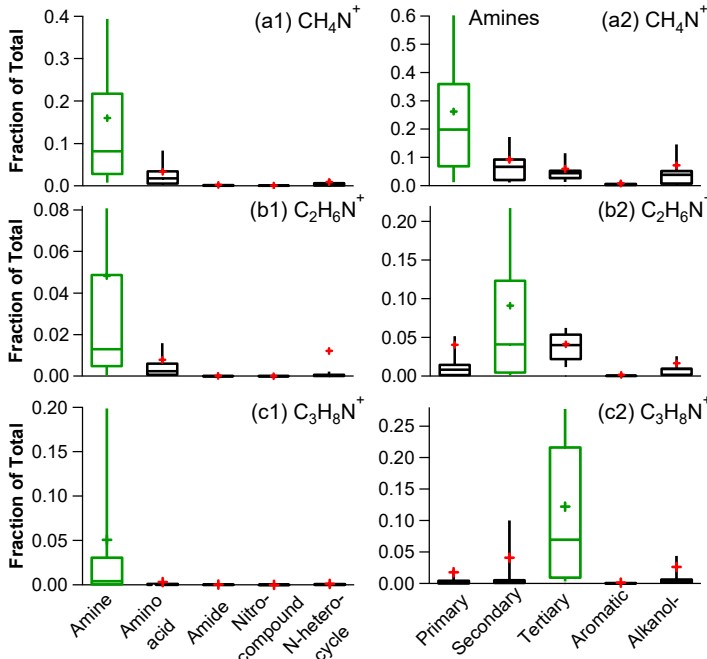

**Figure 3. Fractional contributions of CH₄N⁺ (a), C₂H₆N⁺ (b) and C₃H₈N⁺ (c) to the mass spectra of the N-containing organic species with different functional groups (left panel) and different classes of amines (right panel) (the whiskers above and below the boxes indicate the 90th and 10th percentiles, the upper and lower boundaries of the boxes indicate the 75th and 25th percentiles, and the lines in the boxes indicate the median values and the cross symbols indicate the mean values).**



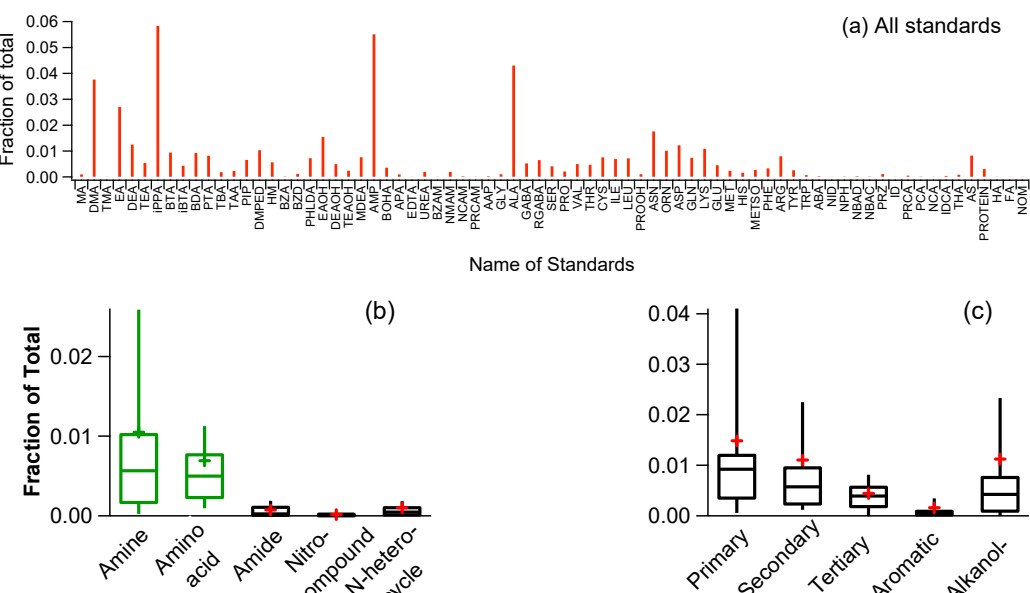

**Figure 4. Mass fractional contributions of NH$_4^+$ in the HR-AMS spectra for all standards (a), different ON classes (b) and different amines (c).**

665

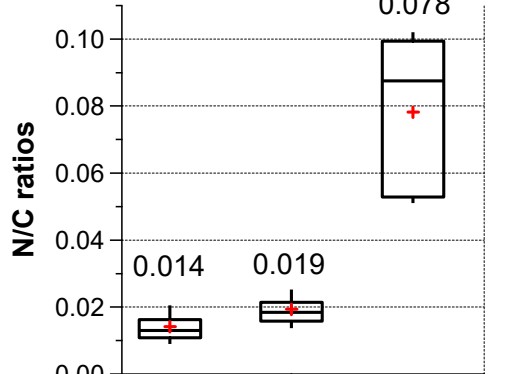

**Figure 5. N/C ratios of the organics in NYC PM$_1$, Fresno PM$_1$ and fog waters (the whiskers above and below the boxes indicate the 90th and 10th percentiles, the upper and lower boundaries of the boxes indicate the 75th and 25th percentiles, and the lines in the boxes**
670 **indicate the median values and the cross symbols indicate the mean values).**



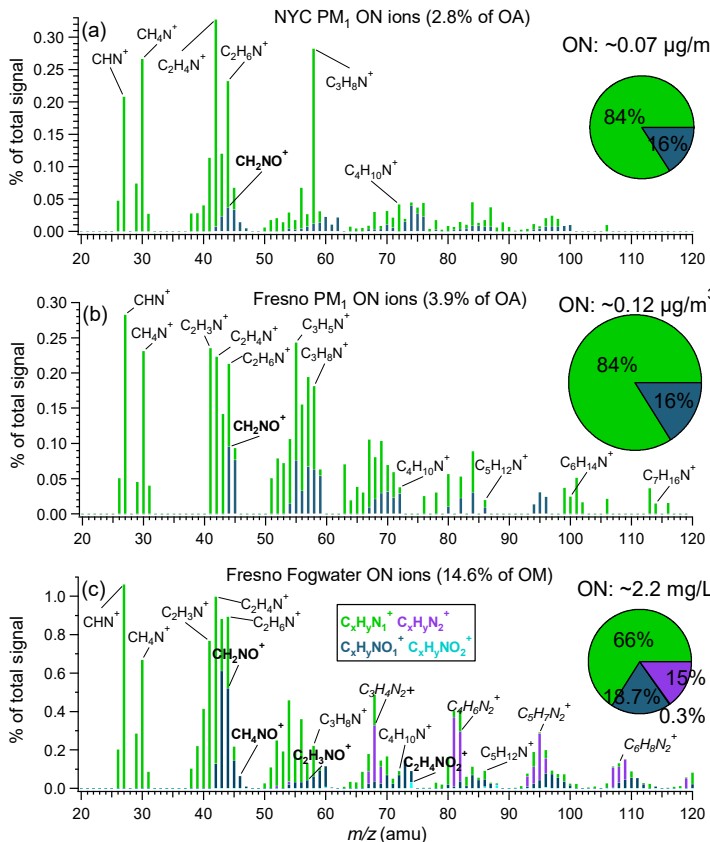

**Figure 6.** Average mass spectra of ON ion categories ($C_xH_yN_1^+$, $C_xH_yN_2^+$, $C_xH_yNO_1^+$, and $C_xH_yNO_2^+$) for NYC PM$_1$ (a), Fresno PM$_1$
(b) and Fresno fog waters (c). The inset pies show the average ON concentrations and mass contributions of the four ion categories
(legends in c) to the total ON, respectively.

675