# Peer review of "Enhancing characterization of organic nitrogen components in aerosols and droplets using high-resolution aerosol mass spectrometry"

_EGUsphere, 2023_

## Author Comment (AC1)

**Response to reviewers**

We thank the reviewers for their thoughtful and valuable comments, and we have incorporated their suggestions into the revised manuscript. Listed below are our point-to-point responses to the comments (in italic) and the corresponding manuscript revisions in quotation marks.

**To Reviewer #1 (Yong Jie Li)**

*The authors presented careful characterization of aerosol mass spectrometric (AMS) measurements of 75 nitrogen-containing organic compounds using high-resolution AMS (HR-AMS). Fragmentation patterns of these atmospherically relevant compounds are discussed and tracer ions are proposed. In addition, based on calibration of 18 nitrogen-containing organic compounds (mixed with ammonium sulfate), an average relative ionization efficiency (RIE) of 1.52 was found to be not much different from the RIE of organic aerosol (OA) in general (1.4); a new calibration factor of 0.79, however, is recommended for N/C estimation for ambient datasets, instead of the widely used 0.96. After the methodology was established, it was used to evaluate the nitrogen-containing organic compounds in two PM1 dataset (Fresno and NYC) and one fog water data set (Fresno), with both concentration and N/C ratios quantified. Finally, limitations of the methodology are also discussed. This is a well designed and rigorously conducted experiment that gives very useful methodological essence in using the HR-AMS to quantify nitrogen-containing organic compounds in the atmospheric aerosols and waters. The manuscript is also very well written with clear demonstration of results and conclusions. I therefore recommend Minor Revision with a few comments as follows.*
*Specific:*
**Comment #1-1:** *Please clarify why organic nitrates (and nitriles, as the authors stated in P8/L240) are not used in this study.*

**Reply #1-1:** Thank you for your comment, which echoes a concern raised by reviewer #3. We have taken steps to address this issue by explicitly stating in our manuscript that our work excludes organic nitrates, and by using the term "non-organonitrate organic nitrogen (NOON)" throughout the manuscript, as suggested by reviewer #3. We also outline the reasons for excluding organic nitrates in the revised manuscript:

(1) The fragmentation of organic nitrates ($RONO_2$) in the AMS predominantly generates $NO^+$, $NO_2^+$, and organic moieties ($R^+$) that lack nitrogen. These ions cannot be used as distinct fingerprint ions for identifying $RONO_2$ from the high-resolution mass spectra (HRMS).

(2) The quantification of organic nitrates can be estimated by using different $NO^+/NO_2^+$ ratios from organic nitrates and inorganic nitrate (ammonium nitrate). This methodology, established by Farmer et al. (2010) and improved by Day et al. (2022), has been widely adopted in numerous AMS studies (e.g., Huang et al., 2021; Yu et al., 2019; Xian et al., 2023). We have offered a general overview of this approach in the manuscript. A detailed characterization of organic nitrates falls beyond the scope of this study.

(3) Additionally, a practical limitation in our work is the lack of representative organic nitrate and low volatility nitrile standards. Nevertheless, nitriles are typically present in trace amounts in ambient aerosols. Thus, our research primarily focuses on common N-containing organic species in the atmosphere such as amines, amides, and others.

*Comment #1-2: Please clarify how the compound class was categories if a compound contains both functional groups, for instance compounds #31, #32, and #60 in Table S1.*

**Reply #1-2:** Indeed, there are a few compounds that contain multiple functional groups. In such instances, we categorize the compound based on its most prominent or reactive functional group, which is often determined, in part, by its naming conventions. For example, compounds like nicotinamide and pyrazinecarboxamide are classified as amides.

*Comment #1-3:P8/L250: it would be good to put a row of fractional signals of averages/standard deviations for M^+ in Table 1 to echo the discussion here, such that readers have a broad idea on what compound classes have higher molecular ion contribution without refereeing to the SI. Although it does not align well with the mass contribution of nitrogen-containing ions to total organic nitrogen mass, it would be good to have such information easily referred to. Better yet, it would be good to make another section in the table to show the fraction signals of main nitrogen-containing ion (and M^+) to the overall mass spectra.*

**Reply #1-3:** As suggested, we have now added a row to show the average fractional contributions of $M^+$ (molecular ion) with one standard deviation for different types of NOON species in Table 1. It should be noted that, the fractional contributions of $M^+$ differs greatly among different ON species, even for the same NOON type, thus the standard deviations are large. The relevant discussions have been rephrased to emphasize those with large $M^+$ contributions.

*Comment #1-4:P11/L325: the bimodal size distributions of ON in marine aerosols here might be linked to the difference between Fresno PM1 and fog water results discussed right above it. It would be good to have one sentence to make the linkage.*

**Reply #1-4:** This is discussed in Section 3.6, and to establish a link, we have added the following sentence: "As previously discussed in Section 3.6, the difference between Fresno $PM_1$ and co-collected fog water could, in part, be attributed to this limitation."

*Comment #1-5:Technical:*
*P6/L193: I do not see it necessary to use "respectively" here.*
*P8/L226: "Additional" to "Additionally"*
*P8/L249: "- a simple amide" to "(a simple amide)"*
*P9/L274: Change this "Furthermore" to "In addition"?*
*P9/L279: "70 ev" to "70-eV", also in P11/L335.*

**Reply #1-5: Revised.**

**To reviewer #2**

*This paper employs high-resolution aerosol mass spectrometry (HR-AMS) to provide an exhaustive characterization of organic nitrogen compounds within the atmosphere. The study encompasses an analysis of 75 distinct types of organic nitrogen standards. The authors not only explore the mass spectral features and N/C correction factors but also propose identifiable ion series, which prove invaluable in facilitating the speciation analysis of organic nitrogen. Furthermore, the authors extend the utility of this method to investigate three environmental samples, analyzing the contents and compositions of organic nitrogen across various regions and phases. This is an important paper for the AMS community. However, there are some minor issues that need to be addressed before publication:*

*Comment #2-1: I would suggest to make a table summarizing all the previous AMS measurements of the ON standards. In this way, the novelty of this paper would be obvious.*

**Reply #2-1:** Thanks for the suggestion. We have conducted an extensive search for published literature detailing AMS measurements on ON standards. In fact, such studies are quite scarce. Therefore, we think it is appropriate to provide a concise summary in the introduction, rather than presenting the information in a table. Notably, the work by Aiken et al. 2008 includes the highest number of ON standards, but it does not report ON spectra. Those spectra data are reported in this study.

The revised texts read: "While the interpretation of OA behaviors has often relied on the O/C and H/C ratios (Aiken et al., 2008; Canagaratna et al., 2015), the N/C ratio has received limited attention, and previous AMS measurements on ON standards are scarce. The work by Aiken et al. (2008) includes the data from 27 ON compounds; however, ON spectra were not reported. These spectra are reported in this study. Ge et al. (2014) introduced a method using HR-AMS to characterize amines and their degradation products in postcombustion $CO_2$ capture (PCCC) processes. In that study, we performed analysis of the AMS spectra of 12 amino compounds and NIST spectra for 37 ON compounds, all of which were identified as the degradation products from PCCC amines. In a separate study, Price et al. (2023) examined the relative ionization efficiencies (RIEs) of three ON compounds. Additionally, Farmer et al. (2010)and Day et al. (2022) proposed AMS-based methods for quantifying organonitrates (aka organic nitrates), which have been widely adopted in various AMS studies (e.g., Huang et al., 2021; Yu et al., 2019; Xian et al., 2023; Lin et al., 2021; Zhu et al., 2016)."

*Comment #2-2: The introduction does not address the limitations of HR-AMS technology. These limitations include an inability to detect substances with low volatility or ionization, differentiate between isotopes, and provide structural insights. To enhance the paper's quality, it is suggested to explain these constraints in the method or discussion section and assess their impact on the obtained results.*

**Reply #2-1:** AMS uses a vaporizer maintained at ~ 600ºC to volatilize particles under high vacuum conditions, enabling the measurements low volatility compounds. It uses 70 eV electron impact for ionizing molecules, which serves as a universal ionization technique capable of ionizing virtually all molecules. Furthermore, it is well known that the 70 eV ionization process induces repeatable fragmentation, providing valuable insights into the chemical structure and bonding of the molecules. Thus, AMS mass spectral data do provide structural insights. In addition, the HR-AMS can differentiate isobaric ions, including isotopes.

Nevertheless, in response to this comment, we have expanded our discussion of the limitations of AMS in the "Method limitations" section. For example, using an average RIE for ON compounds may lead to underestimation or overestimation of specific ON compounds with RIE values largely different than the average value. Extensive fragmentation of molecules induced by the 70 eV ionization process cause difficulties in identifying individual ON compounds. The revised texts read "The AMS's thermal vaporization at ~600°C limits the detection of non-refractory species, potentially resulting in the underestimation of $ON_{NOON}$ levels if the amount of refractory N-containing species is significant. It is worth noting that the contribution of refractory ON to total ON is not yet well understood. Additionally, precise quantification of $ON_{NOON}$ content faces challenges due to relatively significant uncertainties in the N/C correction factor and RIE values. Furthermore, the 70 eV ionization process leads to extensive fragmentation, which adds complexity to the identification of individual NOON compounds."

*Comment #2-3: The paper does not consider the possible mixing effects when analyzing the fingerprint ion series, namely, different types of ON may coexist in the same sample, resulting in signal superposition or interference. It is suggested to discuss this situation in the discussion section and provide methods to distinguish or resolve it.*

**Reply #2-3:** It is true that when multiple types of ON coexist in samples, the association between specific ions and individual molecules or compound classes can get complicated, leading to increased uncertainties in assigning them to particular ON types based on HR-AMS spectra. This challenge has been discussed in section "4. Method limitations", and the revised text is as follows:

"The current method provides insights into ion fragments (fingerprint ions) associated with specific ON functional groups. However, the extensive fragmentation induced by the 70 eV electron ionization in HR-AMS introduces inherent uncertainties in such identifications. For example, while a specific ion like $CH_4N^+$ likely originates from amines, we cannot rule out the possibility (albeit low) of its association with other ON types. In real atmospheric samples containing multiple ON varieties, the overlay and interference of specific ion fragments can be significant, increasing the level of uncertainty. To mitigate this challenge, the incorporation of data from soft ionization mass spectrometry techniques (Wang et al., 2019; Song et al., 2022; Mao et al., 2022) alongside HR-AMS data can significantly enhance the analyses of NOON species, thereby reducing uncertainties and advancing our understanding of their composition."

*Comment #2-4: Mixtures of nitrogen-containing organic compounds and sulfate were combined in a 1:1 mass ratio. Why using this mass ratio? Will mass ratio affect RIE of different compounds?*

**Reply #2-4:** We employed a 1:1 mass ratio for all 18 mixtures due to practical reasons, as it simplifies the calculation of the RIEs of ON compounds through AMS quantification of ON species and sulfate mass. The variations in mass ratios should have a negligible impact on RIEs, as RIEs are primarily dependent on the chemical nature of the compounds, rather than their quantities. As a supporting evidence, Canagaratna et al. (2007a) demonstrated a close match between AMS-resolved sulfate/nitrate mass ratios and the actual sulfate/nitrate mass ratios in the particles, with a slope of 1. Furthermore, a recent study by Niedek et al. (2023) showed that the organic/sulfate mass ratio measured by HR-AMS agrees well with the expected mass ratio in the liquid samples across a broad range of ratios. Accordingly, we anticipate a similar correspondence between ON compounds and sulfate.

*Comment #2-5: How did the authors measure fog water with the AMS? I cannot find any experimental details. Did the water be atomized and then dried? Please provide these important details so that other can replicate these experiments.*

**Reply #2-5**: In the original manuscript, we referred to a prior publication concerning the analysis of fog water samples. We have now provided more details in the experimental section. To answer your question, the fog water was atomized and then dried, following a procedure similar to the one used for analyzing aqueous solutions of ON standards presented in the paper, as follows:

"Additionally, we examined a set of 11 fog water samples collected during January 9-16, 2010, in Fresno. These fog waters were collected using two Caltech Active Strand Cloud water Collectors (CASCC). Immediately after collection, the fog water samples were filtrated with 0.45 μm Syringe Filters (Pall Laboratory) and then stored in pre-cleaned high-density polyethylene (HDPE) bottles within a freezer at -20°C until analysis. The analysis of fog water samples involved the utilization of the same HR-AMS instrument and a similar procedure employed for the NOON standards. More details on fog water analysis can be found in Kim et al. (2019)."

*6. Overall, the experimental section is too brief and lack of many important experimental details. For example, the aerosolization of ON standards is not well described. Again, it is very important to provide all these details so that others can replicate these experiments.*

**Reply #2-6:** The revised manuscript now incorporates more experimental details, as follows:

"The procedures for HR-AMS analysis of liquid samples have been comprehensively documented in prior works (Sun et al., 2010; Ge et al., 2014; Ge et al., 2017; O'brien et al., 2019; Niedek et al., 2023). Each NOON standard was carefully weighted, dissolved in Milli-

Q water, and diluted to a solute concentration of ~20 ppm. A portion of the solution (20-40 mL) was dispensed into a sample vial and then nebulized using a Collison-type atomizer with high purity argon as the carrier gas. Aerosol particles were dehydrated in a diffusion dryer filled with silica-gel, reducing the relative humidity (RH) to <5%, before introduction into the HR-AMS. The HR-AMS was operated at a vaporizer temperature of ~ 600 $^{\circ}$C and was alternated between the highly sensitive V-mode and the high mass resolution W-mode (m/$\Delta$m ~5000). For this study, W-mode data with an extended $m/z$ range extended up to 500 amu was used, aligning with our focus on characterizing the chemical composition. To eliminate the interference of the $N_2^+$ signal on $CH_2N^+$ at $m/z = 28.033$ and $CO^+$ at 28.01, the aerosol generation system was initially purged by atomizing Milli-Q water under argon until the $N_2^+$ signal measured by the HR-AMS reached a sufficiently low level. HR-AMS data for each sample were recorded under stable particle flow conditions, and the high-resolution mass spectrum (HRMS) of each NOON standard was derived from the average of at least three stable runs, each lasting 150 seconds. To account for any potential contamination or background signals, a blank measurement was conducted by aerosolizing Milli-Q water between every two samples, following the same procedure. Typically, the measured signals of Milli-Q water were minimal and had negligible influence on the samples."

**To reviewer #3**

*The authors used an expanded set of ambient-relevant nitrogen-containing organic compounds standards to improve the characterization and quantification methods of submicron organic nitrogen species by the Aerodyne high-resolution aerosol mass spectrometer (HR-AMS). This study provided the improved N/C conversion factor and RIE for quantifying organic nitrogen species (non-organonitrates). The authors also did a detailed examination of the mass spectral feature of standards and ambient samples and proposed tracer ions for identifying different types of organic nitrogen species. This work enhanced our understanding of the performance of the HR-AMS and provided insight into the sources and processes of nitrogen-containing organic compounds in the atmosphere. The manuscript is very well written and easy to follow. I suggest publication after addressing the following comments:*

*Comment #3-1:The study excluded organonitrates, which are also nitrogen-containing organic species, in the analysis. Can the authors comment on that? I suggest the authors clarify the reasons why organonitrates are not included in this method development and perhaps use an alternative term instead of "organic nitrogen (ON)" throughout the manuscript to avoid confusion.*

**Reply #3-1:** Thank you for your comment, which echoes a concern raised by reviewer #1 Yongjie Li. We have taken steps to address this issue by explicitly stating in our manuscript that our work excludes organic nitrates, and by using the term "non-organonitrate organic nitrogen (NOON)" throughout the manuscript. We also outline the reasons for excluding organic nitrates in the revised manuscript. Please see details in **Reply #1-1**.

*Comment 3-2: What were the rough concentrations for the standard solutions? How were fog water samples collected? How were samples stored and prepared before nebulization?*

**Reply #3-2:** We prepared the concentrations of ON standards in aqueous solutions to be ~20 ppm. Information regarding the fog water analysis has now been added in the paper. See **Reply #2-5**

*Comment 3-3: Can the authors explain the importance of RIE? One entire section in the Results and Discussion talks about RIE, and yet it is not clear what this value is and why it is important.*

**Reply #3-3:** In the context of AMS measurements, relative ionization efficiency (RIE) is defined as the ratio of the ionization efficiency of a species to that of nitrate, quantified in terms of mass (Canagaratna et al., 2007b). This correction factor is crucial for accurately quantifying the mass of the specific species in question.

*Other comments:*
*Comment 3-4:Line 19, a calibration factor of 0.79 is for what?*

**Reply #3-4:** It is for NOON quantification. Corresponding text has been updated.

*Comment 3-5:Table 1, why are some numbers in bold? Please also clarify what does "± xx" represent*

**Reply #3-5:** Numbers in bold emphasize relative large and significant values, while "± *xx*" denotes one standard deviation. We have included these explanations in the caption of Table 1.

*Comment 3-6:Figure 1, please explain how is the "Avg. Err" calculated.*

**Reply #3-6:** The average error represents the mean of the absolute relative deviations of all samples from the red fitted line in the corresponding figure. This information has now been added in the caption of Figure 1.

*Comment 3-7:Figure 2, do amino acids show CxHyO+ ions in the MS? Why only present the one or two ion families for each species?*

**Reply #3-7:** The mass spectra of amino acids show the presence of $C_xH_yO^+$ ions. It is worth noting that in Figure 2, we have chosen to show only the characteristic fragments that can serve as fingerprint ions for the corresponding ON species. This approach effectively showcase and highlight the crucial spectral features. For a comprehensive view of the full HRMS of all compounds, please refer to Figure S3. We have included this clarification in the figure caption.

[revised manuscript text omitted]